# Blue Light Blocking Treatment for the Treatment of Bipolar Disorder: Directions for Research and Practice

**DOI:** 10.3390/jcm11051380

**Published:** 2022-03-02

**Authors:** Ioanna Mylona, Georgios D. Floros

**Affiliations:** 12nd Department of Ophthalmology, Aristotle University of Thessaloniki, 541 24 Thessaloniki, Greece; milona_ioanna@windowslive.com; 22nd Department of Psychiatry, Aristotle University of Thessaloniki, 541 24 Thessaloniki, Greece

**Keywords:** bipolar disorder, blue-blocking glasses, amber glasses, mania, bipolar depression

## Abstract

Recent results from a small number of clinical studies have resulted in the suggestion that the process of blocking the transmission of shorter-wavelength light (‘blue light’ with a wave length of 450 nm to 470 nm) may have a beneficial role in the treatment of bipolar disorder. This critical review will appraise the quality of evidence so far as to these claims, assess the neurobiology that could be implicated in the underlying processes while introducing a common set of research criteria for the field.

## 1. Introduction

### 1.1. Definition of Blue Light

Light is by definition the electromagnetic radiation that corresponds to the perceptual limit of human eyesight. Light, being a form of electromagnetic radiation, is transmitted as a wave of energy particles from its source to the receiver, the human eye. A beam of light has a specific frequency, wavelength and energy. Frequency is measured in hertz (Hz), with one Hz corresponding to a wave passing a fixed point per second. The distance between two corresponding points of two consecutive waves is the wavelength and it is measured in meters. The energy that propagates with a particular beam of light is measured in photons, a hypothetical particle that corresponds to the smallest quantity of light energy at the given wavelength. The energy of a photon is thus variable, proportional to the frequency of the beam of light, inversely proportional to its wavelength, and expressed in electron volts (eV).

Light ranges in wavelength from about 400 nm up to 700 nm. The perception of wavelength by the human eye offers the brain the ability to discern different wavelengths by the perception of color. A beam of light with a single frequency is thus referred to as ‘monochromatic’. The upper limit of 700 nm is typically perceived by the brain as the color red while the lower limit of 400 nm is typically perceived as the color violet. The perception of blue corresponds to a wavelength of 450 nm, a frequency of 6.66 × 10^14^ Hz and 2.48 eV.

Early research on retinal damage by radiation has determined that lower wavelengths carry a larger risk of damage and are those implicated in cases of solar retinitis and eclipse blindness [1]. The advent of LED lightning has caused significant concern since ‘white LED’ has an intense emission in the blue wavelength, absent in the normal daylight spectrum and linked with retinal damage in experimental models [2]. Blue-light blocking (or ‘amber’) glasses have been proposed as a remedy for users of LED devices who are exposed to this radiation for a prolonged amount of time, in order to prevent any long-term retinal damage but also in order to remedy sleep difficulties that have been linked to the use of LED lighting [3]. Recently, following empirical findings [4,5,6,7], a few clinical studies have focused on the therapeutic potential for blue-light blocking in bipolar disorder (BD) [8,9], reporting promising results. The aim of this critical review is to assess the limited clinical data that have been reported, present the neurophysiological paths that may be implicated and suggest directions for future research and clinical practice. We will examine the potential mechanisms suggesting that circadian rhythm setting may be associated with the pathophysiology of bipolar disorder from an evolutionary viewpoint, hypothesize as to how these mechanisms may be amenable to blue light treatment and the potential confounders, and offer suggestions as to the more effective planning of such treatment in a clinical setting.

### 1.2. Physiology of Color Vision and the Role of Blue Light

Blue light therapy is rooted in the basic physiology of human vision, with the human eye being able to discern between different wavelengths while at the same time these differences are key for a number of important neurophysiological responses. The evolutionary advantage of color perception is considered as high, with the camera-style eye involving gradually to its role we are accustomed to in the span of a hundred million years and color vision arising in early agnathan vertebrates over 500 million years ago (mya) as a tetrachromatic system [10]. The initial role for a separate organ most likely however was that of circadian regulation through basic light sensitivity in the primitive retina, around 600 mya. The first role of the eye would be to accommodate non-image forming vision, i.e., related to circadian regulation, controlling the pupil size, and other functions such as acute behavioral responses to light in rodents [11,12]. A gradual improvement of capability included a fine gradation in photosensitive cells with the evolution of different types of light-sensitive proteins (opsins) [13] and the insertion of retinal bipolar cells between the photoreceptors and the projection neurons. These evolutionary improvements imparted significant function to the primitive eye: discerning of the environmental cues and enhancing a map of the surroundings aided by tactile, sound and olfactory stimuli. The added function necessitated a corresponding increase in neural circuitry that would integrate all the sensory modalities and create a consistent map of the outside surroundings. With the perception of color making possible the discerning of foes from the surroundings, color vision was a major evolutionary advantage for any primate in the constant struggle for survival.

Meanwhile, the perception of light as a regulator of the circadian rhythm was also fine-tuned throughout evolution to include the new capability of detecting separate wavelengths of light. A recent discovery of the circadian photopigment melanopsin highlighted that the circadian rhythm may function independently of light sensitivity that was, up to then, attributed exclusively to retinal rods, or color sensation that is regulated via retinal cones [11]. The new understanding was that intrinsically photosensitive retinal ganglion cells (ipRGCs) that were thought of only as second-order neurons, additionally had a significant role as photoreceptors due to their melanopsin. However, color sensation has an important fine-tuning role; the cycle of natural lighting starts with lower wavelengths in the morning, increases to higher wavelengths during mid-day, then drops again to lower wavelengths in the evening. Around mid-day, blue light is more prominent in the environment and this period corresponds with the period of maximum awareness in humans, an evolutionary choice for our particular species [14]. The specialized cells in the retina that effect color perception and light sensitivity thus have a dual role in fine-tuning circadian rhythms while the melanopsin functions in a general sensory role for intensity shifts in ambient light [15]. The complete circadian photoentrainment, in turn, affects a number of important functions such as melatonin release, body-temperature regulation and feeding behavior.

### 1.3. Blue Blocking Glasses and Bipolar Disorder: The Working Hypothesis

In those few studies that have so far employed BBG for BD [7,8,9] but also for the limited number of studies researching their effectiveness in depression [16,17,18], the patients were instructed to wear the BBG in the evening. Initially, in a 2008 study, the purpose was to mimic the effect of total light deprivation during the evening, an approach that was hypothesized to mirror the approach of light exposure as therapy for depression [7]. The author hypothesized that a broad range of conditions including rapid-cycling BD would respond to this treatment, although the effect would be mostly limited to patients who responded with reduced sleep-onset latency to the treatment, presumably a sub-population in BD patients. The employment of BBG could tacitly imply that BD is in itself directly linked with an abnormal circadian rhythm setting and using BBG somehow resets this abnormality, which in turn has a beneficial result for the patient. Since the first study, the modern-day environment now has excess sources of blue light during the evening, as mentioned above. Thus, all BD patients who presumably are more prone to the disruption of circadian rhythms would be shielded by a potentially negative effect. We will examine the research data on the effects of circadian-rhythm disruption on BD but also whether the existence of BD can be linked to an abnormal circadian rhythm setting and what the best setting for research and practice would be.

### 1.4. Possible Effects of Circadian Rhythm Disruption on Bipolar Disorder

As stated earlier, the genes that are related to image-forming and non-image forming vision convey a large evolutionary advantage; gene mutations that would lessen the effectiveness of vision would be discarded while mutations that enhance it would be favored [19]. Therefore, evaluating the findings of studies on the effect of blue-light blocking on bipolar patients from an evolutionary point-of-view would include the potential benefit that a deviation from normality in non-image forming may offer. It is important to remember that while cases of bipolar disorder may provide us with the clear indication of an underlying dysregulation, milder expressions of the affected genes could be associated with an evolutionary advantage [20]. A study of the association of circadian genes with mood disorders [21] found that while two specific genotypes (rs10462028 in the circadian locomotor output cycles kaput (CloCK) transcription factor gene and rs17083008 in the vasoactive intestinal peptide (VIP)) were associated with BD, there was a reduced susceptibility for BD with the subjects heterozygous for rs10462028 in CloCK and rs17083008 in VIP; this is typical for a mutation that could yield benefits in heterozygous populations but be catastrophic in homozygous ones. CloCK is among the most important core clock genes that are part of the mammalian molecular clock mechanism itself, a self-sustaining pacemaker operating with a periodicity of approximately 24 h to maintain rhythms for most if not all biological processes [22,23]. VIP-expressing neurons in the ventrolateral ‘core’ of the SCN receive synaptic inputs from ipRGCs, and the response of these neurons is thought to be important for maintaining synchrony within the suprachiasmatic nucleus of the thalamus (SCN) [23]. The CloCK neurons in the SCN are reset via the light-induced release of glutamate from retinal projections. Multiple afferent and efferent projections connect the SCN with other cortical centers, resulting in both excitatory and inhibitory effects. Abnormal patterns of excitation in BD that correlate with SCN input include sleep disruption, general behavioral exaltation in response to stress, aggressiveness and risk-taking.

#### 1.4.1. Sleep Disruption in BD and Circadian Rhythm

Reduced sleep duration, latency and need for sleep have been associated not only with manic episodes in bipolar disorder but also with euthymic states. The study of sleep states in bipolar patients has a significant number of important confounders: duration of illness that typically correlates with loss-of-function and employment, persistence of subthreshold depressive symptomatology between episodes which would decrease activity and increase sleep duration and antipsychotic medication that is associated with hypersomnia [24,25]. In this respect, studies of non-affected offspring would provide a clearer picture, provided that the offspring are not themselves affected by a mood disorder. This presumption is easier to satisfy in younger ages. Such a study that compares young, unaffected offspring of bipolar patients to controls has demonstrated decreased physiological catch-up sleep on non-school days [26]. A reduced need for compensatory rest could be considered to offer a slight evolutionary advantage. The authors proposed that this decreased need for sleep may represent an endophenotype of BD.

#### 1.4.2. The Effects of Stress on Circadian Rhythm in BD

The circadian rhythm is also integral to the release of cortisol, the principal stress hormone. The cortisol peak precedes the activity period, and for diurnal humans the effect of light in the environment is to trigger cortisol release in order to prepare us for daytime activity, since it instructs the organism on how to respond to the stress through multiple effectors while influencing a host of neurocognitive processes relevant to fear, learning, and coping [27]. The unaffected offspring of BD patients exhibited increased daytime secretion of cortisol in a related study that measured cortisol for two weeks [28]. A separate study of the effects of chronic stress found that offspring of BD patients who were experiencing chronic or acute stress responded with higher cortisol secretion following awakening than those who did not [29]. The immune response is also affected by the circadian rhythm, with peak response during the rest period [30], mediated by complex epigenetic mechanisms that affect all aspects of immunity [31]. The relationship between emotional states and immunity is a complex one. Changes in immune state can be translated by the nervous system to a corresponding change in emotion, while distinct emotional states (e.g., anger) have been demonstrated to bring about alterations in the immune function [32]. Potent mood stabilizers, such as lithium and sodium valproate, exhibit a protective effect against the increased tendency for apoptosis in the lymphocytes of BD patients [33]; neither these effects on the cellular immunity in BD have been researched for potential linkage to overt psychiatric symptomatology, nor has the abnormal immunity in BD patients conclusively been linked with the hallmarks of the disease. There are considerable differences in immunity among different states of BD patients, with patients in depression and remission showing different ratios of B cells and various interleukins [34], further complicating the issue. Again, research on non-affected adolescent bipolar offspring may be a safer indication of potential evolutionary gains; a related study demonstrated a distinct pattern of increased and decreased levels of various immune growth factors [35].

In all these instances, there are two powerful moderators (‘zeitgeber’) other than light in the external environment, namely stress and social cues to time of day. Stress and the circadian clock system have a complex interaction whereby stress resets and destabilizes the system if it persists long-term in an unpredictable manner, while a disruption in the circadian clock system may act as a stressor in itself [36]. The social zeitgeber theory proposed that the regularity of social time cues, such as bedtime, mealtime, work shifts and exercise play an important role in the entrainment of the circadian rhythm, and although supportive data remain limited, it has provided a backdrop for examining the reciprocal relationship between lifestyle routine and BD [37]. There is experimental evidence to date that irregularity in social-time cues is a risk factor for a new BD episode [38] and that its amendment can benefit BD patients [39]. The onset of disruption in social behavior that is related to the onset of a manic episode (subjective need for sleep, food and rest) is difficult to ascertain and it may in fact precede the onset of full-blown symptomatology [40]. These findings point to the possibility that an innate susceptibility to BD may relate to deficits in the functioning of the epigenetic clock mechanism and these deficits may be amenable to external manipulation. While researchers have attempted to discern between different types of episodes based on sleep-disruption patterns in BD, it appears that sleep disruption is a common feature in manic, depressive and euthymic states [41], a finding congruent with the hypothesis of a pre-existing deficit in the clock mechanism that is expressed throughout the clinical course of BD patients. A related genetic study [42] identified thirteen phenotypes, of which 12 were heritable, that included later awakening, longer duration of sleep, while these euthymic BD patients displayed lower activity levels than non-BD patients. During the active phase, these euthymic BD patients had fewer total minutes scored as awake and more variability in the total minutes scored as asleep. These chronotype aberrations in BD appear as a disease trait and remained significant after accounting for lithium and antidepressant medication. One of the first case reports noted that the patient, a strong responder to the treatment with BBG, had a definite seasonal pattern in his manic episodes [4]. Interestingly, the patient had not responded to lithium treatment in the past. This seasonal pattern is a plausible factor since the amount of evening light varies greatly with season change. It is an element that can be easily extracted from patient’s history and has not been reported in larger studies so far.

#### 1.4.3. Aging and Circadian Rhythm in BD

Research in aging of BD patients offers further concrete evidence that a modest disruption of an innate epigenetic clock mechanism occurs. In fact, mania-like behavior was experimentally induced by a mutation in the circadian locomotor-output cycles kaput (CloCK) transcription factor gene more than fifteen years ago [43]. CloCK was proven to act as a central transcriptional activator of molecular rhythm, while the effect of its mutation was countered with the administration of lithium. There were two possibilities, one that disruptions in the clock mechanism in itself results in a cascade of adverse effects in mood, or an alternate hypothesis that BD may be conceptualized as a disease of accelerated aging characterized by epigenetic changes, including DNA methylation, non-coding RNAs and various kinds of histone modifications [44]. A recent review of the research on the aging hypothesis suggested that multiple biological systems interact with environmental triggers to induce premature aging in BD patients, based on correlations of epigenetic markers found in aging and BD [45]. Although this conceptualization may be easy to follow since it is based on the similarity of some pathology between aging and BD, it would be misleading as a research framework. In fact, the similarities found in both processes may be the common results brought upon the disruption in the rhythm of circadian oscillators in the cell, the disruption itself not being necessarily of the same type or origin. In the above review, the concrete results were somewhat disappointing for the broader proposed commonalities [45]; a reduced telomere length was not conclusively linked with BD patients, with multiple confounding factors distorting the results, most notably general stress levels. Age-related oxidative stress and levels of inflammation have only been shown to increase in late stages of the disease and manic phases, again confounded by multiple other factors and without any conclusive evidence in favor of a common link. Methylation of DNA and the mitochondrial DNA copy number at a given time heavily depended on levels of oxidative stress and were confounded by history of childhood trauma and psychosocial stress.

#### 1.4.4. Monoamines and Circadian Rhythm in BD

Research on the direct effects of disruption of CloCK functioning has shown promise for a better understanding of the link between the expression of BD and circadian-rhythm disruption. Recently, a series of experimental studies on knock-down mice models of mania revealed additional pathways to BD pathology outside the realm of CLoCK mutations. These include a possible role of altered glutamatergic signaling in the neurons of the eyes’ afferent light pathways to the SCN [46] and also diminished CLOCK protein function, thereby increasing dopaminergic activity and tone and altering excitatory drive onto the GABA-medium spine neurons in the nucleus accumbens [47]. The dopaminergic system is heavily engaged in some of the core symptoms of BD, including hyperactivity, increased anxiety, increased aggression and risk-taking and addictive tendencies. Research data suggest that the dopaminergic system is under the influence of the circadian clock [48] and drugs that interact with the dopaminergic system may exert powerful mood enhancement (cocaine and other drugs of abuse [49,50], dopaminergic antidepressants [51], stimulants for ADHD [52]) or antimanic effect (antipsychotics) [53]. CLoCK mutant mice have an increase in dopaminergic activity in the ventral tegmental area, leading to a greater preference for rewarding stimuli, similar to BD patients in mania. Τhis behavior is reversed by expressing a functional CLoCK protein via viral-mediated gene transfer specifically in the ventral tegmental area [43]. This potential link of circadian dysregulation with the dopaminergic system is a candidate for evolutionary benefit that would weigh in favor of genetic survival of the mutations, since hyperactivity, risk-taking, heightened arousal and aggression may incur a slight benefit for the subject in instances of extreme environmental stressors that would be life-threatening. Again, this advantage would manifest in a more productive manner in a moderate form that could be traced in unaffected siblings of BD patients. A recent meta-analysis of psychopathology in siblings of BD patients that included seventeen studies [54] confirmed that they had a significantly increased risk of developing ADHD, substance-use disorder, conduct disorder and oppositional-defiant disorder compared to controls. A separate study of unaffected siblings of patients with BD showed that they displayed the tendency to make poorer decisions compared with normal controls due to negative affective bias associated with impulsivity and risk-taking, these results being similar to those of euthymic BD patients [55].

While dopaminergic activity is linked with circadian rhythm function, there were few data regarding serotoninergic activity and circadian rhythm until recently. Initially the monoamine hypothesis of bipolar disorder claimed a functional excess or dearth of monoamines would lead to manic or depressive symptoms. The circadian dysfunction in this context would lead to either reduced or increased serotonin transmission, which would in turn trigger the corresponding symptoms. However, the situation appears more complicated, since the SCN also contains one of the densest serotonergic terminal plexi in the brain. A related review concluded that the apparent role of serotonin is to modulate the sensitivity of the circadian rhythm to light [56], which in turn would lead to abnormal function. A role of serotonin in depression is also put forward in another review that postulates that stress-induced perturbation of the serotoninergic system disrupts circadian processes and increases susceptibility to depression [57]. Thus, antidepressants that rely on serotonin reuptake inhibition may also affect the circadian rhythm and their use should be taken into account.

#### 1.4.5. Melatonin and Circadian Rhythm in BD

Melatonin is another key neurochemical modulator of the circadian clock. Melatonin is synthesized by the pineal gland during nighttime and serotonin is one of the intermediary substances during its production cycle. While a number of studies on melatonin in BD patients have been carried out, the results remain inconclusive, possibly due to measurement issues [58] or simply due to the fact that melatonin is regulated by downstream processes in the circadian system that may already be at fault (i.e., abnormalities in CLOCK). A genetic study of the melatonin biosynthesis pathway in patients with bipolar disorder, however, proposed an interesting hypothesis [59]. A key enzyme of the melatonin biosynthesis is acetylserotonin O-methyltransferase (ASMT) and results showed generally lower ASMT enzymatic activity observed in patients with BD compared with controls, while several ASMT mutations identified in patients were associated with low ASMT activity. The authors concluded that rare and common variations in ASMT might play a role in BD vulnerability and suggest a general role of melatonin as a susceptibility factor for BD. Since melatonin as a supplement [60] and melatoninergic antidepressants (agomelatine) [61] have been suggested as possible treatments for bipolar depression, this should also be taken into account when assessing patients in a future study. Measuring melatonin levels should also be considered during a future study, since a better regulation of the circadian rhythms could act as a downstream regulator of abnormal melatonin secretion.

## 2. Materials and Methods

Initially, a brief literature review was conducted in the PubMed/MEDLINE and Scopus databases with the following keywords: blue blocking glasses OR amber glasses OR blue light AND (mania OR bipolar), yielding 183 results from the past fifteen years. Inclusion criteria were as follows: clinical studies of any type using BBGs with diagnosed BD patients of any BD subtype, in any setting. Exclusion criteria were patients with other psychiatric comorbidities. After filtering out the results according to these criteria, we assessed only a handful of remaining clinical studies [7,8,9] for their treatment of confounder variables, with a limited number of case reports presenting empirical findings [4,5,6,7].

## 3. Results

### 3.1. Research Data So Far

Table 1 presents the studies that were assessed during the literature review in brief.

Out of the three relevant papers [7,8,9] there were only two case–control clinical studies that focused on the therapeutic potential for blue-light blocking in bipolar disorder (BD) [8,9], reporting promising results, since results from a 2008 study were of a naturalistic design, offering outpatients the chance to try out the potential treatment but with no control group [7]. Interestingly, the next study came some eight years later; Henriksen et al. [9] conducted a randomized controlled trial of 42 manic inpatients under pharmacological treatment and concluded that patients who wore blue-light blocking glasses (BBG) had higher sleep efficiency and a rapid decline of symptoms, assessed with the Young Mania Rating Scale (YMRS). Additionally, motor activity was assessed via actigraph recordings and this, along with the YMRS item scores related to increased activation, declined before the YMRS items related to symptoms of distorted thought and perception. This led the researchers to hypothesize that the primary anti-manic effect of BBG is deactivation. The intervention lasted for a week but improvement was evident from day one, with high effect sizes in general and the patients tolerating the BBG well.

Esaki et al. [8] assessed the impact of BBG in euthymic BD outpatients with insomnia who used them for two weeks, from 20:00 to bedtime. In a previous study where the research team examined the effectiveness of BBG against major depressive disorder, 40% of the participants reported pain or discomfort from wearing the BBG for an extended period of time [16]. The authors thus used BBG with a size-adjustable capability in this study. While the authors did not detect and changes in actigraph-recorded sleep parameters, mood symptoms or subjective sleep quality, there was a shift of the circadian rhythm towards the desired direction. However, the results were hampered by the low effect size and concomitant use of antidepressants in the BBG group.

### 3.2. Alternates to Present Research

It is worth noting at this point that there have been suggestions of using a light environment in hospitals that uses blue-light blocking centrally to mimic evening light conditions in order to produce less disruptive effects on the circadian system and improve sleep [62]. The obvious issue with designing a case–control study of the impact of ward lighting is that the patient population is mixed and the segregation of patients from controls is impossible unless they are strictly limited to confined spaces. The only way to resolve this issue is the creation of two units that differ only in the lighting system. An attempt to realise this took place in Trondheim, with close to 500 trial participants who were randomized either to a ward where light sources in all rooms were blue-depleted between 06:30 p.m. and 07:00 a.m. or to a ward employing normal hospital lighting [63]. Results on sleep patterns and melatonin levels were encouraging [64]. This study design is obviously impractical for research but it could be repeated on an outpatient design with appropriate house lighting. BBGs on the other hand can be easily employed both in research and clinical practice in wards and living environments, with the added benefit of being practicable in work environments. Hence, this narrative review will focus more on the use of BBG although all pathophysiological processes that are described also apply to other ways of mimicking an evening light environment.

## 4. Discussion

### 4.1. Quality of Evidence and Potential Confounders

While these results show some progress, there is a lack of a standardized procedure for this treatment, limited examination of potential confounder variables and the clinical potential may be underappreciated. Esaki et al. [8] screened outpatients whose type of BD or current mental status was not considered. Antidepressant use was considered as a confounder variable but there was no mention of mood stabilizer or antipsychotic use, nor was there a measure of sedation attributed to the drug regimen. The patients were instructed to wear BBG after 20:00 p.m. until bedtime. Henriksen et al. [9] recruited inpatients admitted with bipolar mania. Patients were instructed to use the glasses from 18:00 p.m. to 20:00 and were included in the analysis if they did so for at least one evening, night and early morning. The dosages of the medication were reported in detail for each patient along with individual outcome with regard to the Young Mania Rating Scale score. Nearly all of the patients received a combination of antipsychotics and mood stabilizers.

It is clear from these two very early stage clinical trials that there is a multitude of related variables of which the possible effects should be taken into account while assessing the feasibility of using BBG in practice. There are practical issues, such as the actual effectiveness of the BBG units in their stated role to block blue light and patient comfort while wearing these BBG units. However, there also a significant number of key issues regarding patient’s current state that would be appropriate for the treatment, including the type of BD (BD type I, II), phase of the disease (acute episode of mania, depression or mixed-episode, euthymic, hypomanic or dysthymic), and, the assessment of the appropriate time of day for wearing BBG and duration of wear. Finally, numerous potential confounders that include sleep routine, work shifts and current medication regime are relevant factors. These issues are becoming all the more important since the concept of BBG has been widely reported in the media to the extent that there is little chance of running a clinical trial with a placebo group, since BBGs are impossible to mask both from the patient and the researcher due to their characteristic amber hue. To a large extent, these gaps in the literature stem from an approach that is practical but with little linkage to research findings on the underlying psychophysiology.

### 4.2. Guidelines for Research and Practice

#### 4.2.1. Research Setting

Studies so far have included both inpatients in a clinical setting and outpatients. A clinical setting may be more easily controlled and provides the opportunity for a control population who would not wear active lenses. Naturalistic settings involve patients in remission and should be organized during the same seasons and locations that do not vary considerably in latitude for all patients, so as to not have considerable variations in the day–night cycle between patients.

#### 4.2.2. Study Populations

To summarize the very limited experimental data so far, we should note that they point to the possibility of a specific endophenotype of BD patients who are more prone to circadian-rhythm disruption. This should correspond to patients with a specific pattern of disruption in their corresponding genes, one that could provide a slight benefit in reducing the need compensatory sleep in milder, subclinical cases, as was demonstrated in the unaffected siblings of patients with BD. Hence, a first guideline would be to further expand research on unaffected siblings of patients with BD. Although the genetic burden would be variable, the reduced number of confounder variables in unaffected individuals makes it worthwhile. If unaffected siblings are tested early enough in their lifespan, there is a possibility that in due time they could be diagnosed with BD. These data will be of additional importance compared to senior siblings who are not diagnosed with BD.

#### 4.2.3. Type, Phase and Age of Onset of BD

An important parameter in any study of BD patients is the diagnostic consistency of the disease. A recent comprehensive review of published articles found a mean prospective consistency of 77.4% and a retrospective consistencies of 67.6% [65]. The results are acceptable as a whole, yet the inclusion of patients with a first-time episode in a study would be best avoided. The type and phase of BD is most likely of lesser importance in a cross-sectional study but monitoring a patient longitudinally as he/she transitions from a manic episode to a euthymic state would provide useful data in how the BBG affects circadian variability in the long term. So far, data have been available from the acute phase of BD and while they are interesting from a clinical viewpoint, they are fraught with confounder variables, including pharmacologic-treatment modalities. Typically, the BBG are considered to be an add-on treatment and thus patient treatment varies greatly. Refractory-to-treatment patients could have multiple confounding factors at play and could mask the results, including rapid-cycling episodes. A better alternative would be a separate study on patients with rapid-cycling or refractory to treatment-as-usual but in an euthymic state while under study. While a BD type I diagnosis typically has a large percentage of interrater reliability and wide acceptance, this is not the case for a BD type II diagnosis where considerable controversy exists on whether patients should be simply classified as being in a bipolar spectrum [66]. So far, the experimental studies have grouped all BD diagnoses together and no discerning elements have been reported between BD types. It is unclear whether the I and II subtypes would be somehow differentiated by severity of circadian dysfunction, although a depressive episode in both major depression and bipolar depression may be treated with blue-light exposure. More comparative research would be helpful.

Age of onset is an understudied variable. BD has been shown to have either a trimodal (early onset, mid-onset and late-onset) or a bimodal age of onset in a recent review [67]. The average age of early onset was 17.3 years (SD = 1.19) corresponding to 45% of total cases, for mid-onset 26.0 years (SD = 1.72) with 35% of total cases and for late onset 41.9 years (SD = 6.16) in 20% of total cases. For the bimodal distribution, the average age of early onset was 22.5 years (SD = 7.32) with 63% of total cases and for late onset was 40.8 years (SD = 16.89) in 37% of total cases. Regarding age of onset of BD, we cannot at this point in time hypothesize as to whether it would be a parameter that could discern patients who would respond better to BLB treatment. Unfortunately, the experimental studies so far failed to either record this variable or compare responders to non-responders on it. We would, however, expect an earlier age of onset to be predictive of response, since a genetic disturbance would manifest its negative effects early. Additionally, with the preponderance of sources of blue light in our modern-day environment, it is plausible that an additive negative effect could affect patients who would otherwise remain asymptomatic for longer. Thus, sources of blue light in the living environment need to be taken into account and neutralized with appropriate solutions. Thankfully, there is a number of alternatives both for lighting but also for the use of digital screens in general with applications that change the color temperature of screens. Hence, this variable needs to be both researched and controlled for in any given research and clinical setting.

#### 4.2.4. Patient History Taking

A complete patient history should include exhaustive information regarding his/her chronotype, social rhythms and seasonality. Patients should be matched to controls regarding matters that are outside the patient’s control, such as type of work and especially regularity of work shifts while sleep patterns and activity need to be recorded. Seasonality is an element that can be easily extracted from patient’s history and has not been reported in larger studies so far. It should be taken into account when conducting a clinical trial; this may be challenging considering that typical studies may be extended for a span of time that encompasses two seasons or more.

#### 4.2.5. Recording of Patient Pharmacological Treatment

A very important confounder variable is current treatment. As reported above, lithium and, to a lesser extent, sodium valproate, have been linked with circadian-rhythm regulation. The presence of these agents in the treatment regime could mask the full effect of BBG treatment. A study comparing treatment efficacy for lithium between patients who responded to BBG and non-responders may yield interesting data. Antipsychotics and benzodiazepines have not been reported to have meaningful effects in the circadian rhythm. However, given the sedentary nature of a number of compounds, they could indirectly affect the results by affecting sleep and alertness. Antidepressants that affect dopaminergic, serotoninergic or melatoninergic transmission may directly affect the circadian rhythms of patients as well. Discerning between the type of antidepressants and response to BBG may also provide some interesting research data.

#### 4.2.6. Laboratory and Neuropsychological Testing

A comprehensive test battery that can be used to determine the elements in the presumed endophenotype of BD patients with circadian-rhythm dysfunction by examining response to BBG treatment may be a potential research goal. This should include reliable measures of cortisol, melatonin, sleep quality and latency, impulsivity, risk-taking behaviors, arousal, aggression and hostility. Cortisol measurement may be carried out using hair for an up-to-three-month retrospective assessment or saliva to record current variation during the waking day, as carried out elsewhere [68]. Saliva melatonin measurement can be a useful supplement to saliva cortisol measurement [58]. While these elements are negatively affected in all BD patients, it would be interesting to see whether there are meaningful differences between BBG responders and non-responders. If BBG responders have worse scores than non-responders, it is possible that causation flows from the direction of circadian dysregulation to the general dysfunction of dopaminergic regulation and is not a mere correlate. The issue of cost and scalability comes into play. Administering a neuropsychological test battery to assess behavioral aspects is a relatively easier process and does not require a sleep laboratory. Measuring cortisol, melatonin or/and dim light melatonin onset are more expensive but also more exact methods. Actigraph recordings of activity are becoming cheaper with the advent of microtechnologies, with a large number of commercial products already offering some level of activity monitoring. Lately, a number of state-of-the-art machine-learning approaches have been proposed to assess the circadian time from a single blood sample [69,70]. Researchers will soon be offered new tools with great potential.

#### 4.2.7. Type of BBGs

Finally, there is a practical element that should not be overlooked. The clinician should bear in mind that not all blue-blocking glasses are created equal. Amber lenses may have the higher potential for blocking blue wavelengths and are useful for outdoor use or for use in a controlled clinical setting but they cannot be employed indoors and they skew the perception of color, leading to potential personal safety risks. Most commercially available blue-blocking glasses are tinted amber but also attempt to preserve color perception, which leads to compromises. A recent survey [71] of seven commercially available blue-blocking glasses found the actual reduction ranging from 6–43% while circadian sensitivity was reduced by 4–27%. While all glasses provided a degree of protection against photochemical retinal damage, it is unclear as to the actual meaningful impact on the circadian system would be. Circadian sensitivity was calculated as a function of the spectral sensitivity of the ipRGC cells. To date, there is no experimental study that compares the spectral sensitivity of the ipRGC cells of BD patients to controls, thus the optimum degree of reduction is unknown. Lens transmittance alone cannot determine the circadian impact completely, due to idiosyncratic factors (e.g., pre-existing eyesight issues that require the use of corrective lenses and the transmittance of ocular media). In an experimental setting where the subjects were exposed to evening bright-light LED to a resetting of the circadian pacemaker, a mere 1% dosage of light achieved a level of response comparable to 50% of the full dosage [72]. Thus, the potential of using blue-blocking lenses that do not hinder color perception yet impart a meaningful effect on circadian rhythm appears good. A numerical index, the Non-Linear Circadian Index, has been proposed recently to summarize the impact of the transmitted light on the circadian cycle [73]. The authors suggested that further research should be carried out with retinal cells in vitro and we would expand on this suggestion in that additional research should be carried out in the field with BD patients wearing specific lenses during a specific time of day and level of activity. A summary of the main findings can be found on Table 2.

## 5. Conclusions

Research on blue-light blocking for the treatment of BD is still very limited. This in part is due to the limited body of knowledge of the effect of circadian-rhythm disruption on the course of the disease, and the source of this disruption. The ease of application of this novel treatment, however, may be useful in order to ascertain the differential impact of various factors on circadian disfunction in bipolar disorder. Careful planning of future studies is of paramount importance, given the complexity of the underlying pathophysiology.

## Figures and Tables

**Table 1 jcm-11-01380-t001:** Clinical studies involving blue light blocking glasses in bipolar disorder.

Study	Design	Patients	Intervention	Outcome Measure	Main Results	Conclusions
Phelps [7]	Case series	21 consecutive inpatients	Patients received BBG to wear from 20:00 h to bedtime. All had insomnia. Duration of treatment was undisclosed	Clinical Global Improvement Scale (CGI) score	52% of patients improved in CGI, most of them (42%) very much so. 38% did not respond	Results were promising but lack of control or placebo limited their significance
Esaki et al. [8]	RCT	43 outpatients with BD and insomnia divided in two groups	Research group received BBG and placebo group clear glasses to wear from 20:00 h to bedtime for 2 weeks	Visual Analog Scale (VAS) self-assessment of sleep quality, Morningness–Eveningness Questionnaire, sleep actigraphy, evaluation of mood	No difference in VAS, sleep actigraphy or mood symptoms, improvement in sleep rhythm	BBG may be useful as adjunctive treatment for BD
Henriksen et al. [9]	RCT	32 inpatients with BD divided in two groups, after drop-outs 12 patients in the research and 11 patients in the placebo group	Research group received BBG and placebo group clear glasses to wear from 18:00–20:00 h for 1 week	Young Mania Rating Scale (YMRS), actigraphy recording of motor activity	Statistically significant difference in improvement of YMRS score in favor of research group	BBG are effective and feasible as add-on treatment for bipolar mania

**Table 2 jcm-11-01380-t002:** Suggestions for future research by research variable.

Research Variable	Suggestion
Setting	A clinical setting may be more easily controlled and provides the opportunity for a control population. Naturalistic settings should be organized during the same seasons and locations that do not vary considerably in latitude for all patients.
Population	Ideally, non-affected siblings of BD patients. If a patient population, then refractory-to-treatment patients and patients with mixed episodes should be examined separately. Include age of onset and duration of untreated and treated disease as confounders if matching on these variables is unfeasible.
Type of bipolar disorder	Bipolar I or II, provided the diagnosis has already been established and the patient has been followed previously to avoid issues with diagnostic accuracy.
Phase	Manic or depressive episode in clinical settings, euthymic in naturalistic settings.
Patient history	Record information on chronotype, social rhythms, seasonality, line of work and work shifts. For patients studied in a naturalistic setting, this information should ideally be unchanged for the duration of the study. Record any changes in this information in clinical patients prior to the latest episode.
Current treatment	Careful recording of type and duration of current treatment. Record separate variables for lithium, valproate, antidepressants and sedating medication. Record previous treatment regime if discontinued prior to relapse and level of response.
Laboratory tests	Cortisol secretion via measurement in hair and saliva, saliva melatonin measurement, activity, portable actigraph measurements of sleep quality and latency; new machine-learning approaches may assess circadian time from blood samples and will become widely available in the near future.
Neuropsychological examination	Measures of impulsivity, risk-taking behavior, arousal, aggression and hostility.
Type of blue-blocking glasses	Amber lenses for a clinical setting, appropriate lenses that do not skew color perception for naturalistic settings. Any type of lenses should be assessed for its effectiveness.

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
