# Peer review of "Blue Light Blocking Treatment for the Treatment of Bipolar Disorder: Directions for Research and Practice"

_jcm, 2022, doi:10.3390/jcm11051380_

Round 1
Reviewer 1 Report
This Manuscript is a Review about blue light treatment und the role of circadian rhythm in bipolar disorder. The general tonic is of Interest. There are some issues that should be addressed before publication.
W. Material und methods: what were the in- and exclusion criteria for the selected Studies?
E. How did the authors define "relevant" Studies to include in the Review?
Diskussion: in the discussion section, a lot of times, reference are missing, for example;
Line 177, Page 4
Line 208, Page 4
Line 211, Page 5
"Iines 234-242, Page 5
Line 248-250, Page 5
Line 298, Page: responders to what treatment?
Line 362, Page 7: References missing
Line 346, Page 7: references missing
Line 393, Page 8: use agomelatine Not the Brand Name valdoxan
Author Response
This Manuscript is a Review about blue light treatment und the role of circadian rhythm in bipolar disorder. The general tonic is of Interest. There are some issues that should be addressed before publication.
Response: Please note that due to drastic restructuring of the manuscript following the suggestions of Reviewer #2, line numbers are changed; however, the changes are highlighted in yellow in the revised manuscript. Also, other major changes are highlighted in yellow; these include both new material and the subheadings of the sections that were moved in the manuscript and not the entire section because the order of the manuscript has been changed dramatically and highlighting all material that was shifted between sections would not be practical.
- Material und methods: what were the in- and exclusion criteria for the selected Studies?
Response: Thank you for your comment, inclusion and exclusion criteria are included in the revised manuscript.
- How did the authors define "relevant" Studies to include in the Review?
Response: Thank you for your comment, the relevant studies correspond to the inclusion/exclusion criteria, this has been made clear in the revised manuscript.
Diskussion: in the discussion section, a lot of times, reference are missing,
Response: Thank you for your comment, all suggested changes were made as follows:
Line 177, Page 4 – inserted reference, now in line 91
Line 208, Page 4 – inserted reference, now in line 124
Line 211, Page 5 – inserted reference, now in line 130
"Iines 234-242, Page 5 – inserted two references now in line 156
Line 248-250, Page 5 – inserted reference now in line 256
Line 298, Page: responders to what treatment? Treatment added now in line 216
Line 362, Page 7: References missing reference was in the next sentence, sentences were joined now in line 279
Line 346, Page 7: references missing: - inserted appropriate references now in lines 262-263
-Line 393, Page 8: use agomelatine Not the Brand Name valdoxan : resolved as suggested now in line 309
Reviewer 2 Report
jcm-1576750: Blue light blocking treatment for the treatment of bipolar disorder: directions for research and practice
This is a failed research paper, at least in the way it is presented. The content of the paper is of great interest, and the overall writing is good. The references that have been analyzed are current and well reviewed.
However, the structure of the review is not clear. It seems more like a non-systematic review of circadian rhythm disruption. In this sense, we propose to the authors to modify the presentation of the contents; under this idea, to modify the title, and in the same, to point out in some way, a special or particular attention to blue light blocking treatment, due to the scarcity of information on the subject.
If, on the other hand, the idea remains the development of a review of blue light blocking treatment for bipolar disorder, the reformulation would be in the direction of a better organized review. The major part of the discussion should be the introduction, aimed at setting the background of the content related to circadian rhythm disruption, its relation to bipolar disorder, the ways to approach it, and the existence of blue light blocking treatment. What is blue light blocking treatment, how is it applied, what would be its mechanism of action, etc.? With this justification, the review could be developed. It should be done in a more correct way, for example, following the PRISMA guidelines, precisely because the lack of rigorous studies on this content can be well justified.
Then, the criteria to be followed for the selection of the articles to be analyzed should be specified. The discussion, even if it is now shorter, should analyze the fundamental aspects, link it to circadian rhythm disruption, point out the drawbacks, its advantages, the position of the authors themselves, and the proposals that the authors make for intervention and research. In fact, what is relevant is to discover, first, what has made it less researched; what is the relationship with light therapy for depression and bipolar disorder, and what steps should be taken from now on regarding this form of intervention.
The importance of the variables to be taken into account in the design of an intervention, how to establish the effectiveness of the treatment, the biases committed by previous research, and the criticisms that the previous authors themselves have established in the treatments, are also necessary aspects in a review due to the scarce information available. A table with a summary of the main key points to be taken into account in the contributions of the articles reviewed would also be helpful.
The idea, therefore, is more of a decision on what to emphasize, the reorganization of the contents, and a clearer, more evident systematization of the analysis carried out. The authors are encouraged in this sense, because the information analyzed is relevant, the content chosen is of interest, and it remains to be seen exactly in what way it can be shown that it is worthwhile to investigate or not in this direction.
Author Response
Response: Thank you for your kind words and your helpful suggestions. The re-organization of the manuscript was been carried out and the manuscript now has a more logical flow for the reader. The introduction section has been expanded to include material previously presented in the Discussion section. The Discussion section is clearly divided in sub-sections on suggestions for research that are also presented in a condensed form as Table 1.
Kindly note that major changes are highlighted in yellow; these include both new material and the subheadings of the sections that were moved in the manuscript and not the entire section because the order of the manuscript has been changed dramatically and highlighting all material that was shifted between sections would not be practical.

Round 2
Reviewer 2 Report
Second revision.
The authors have made a valuable effort to reorder and make the work more understandable. This is something that is appreciated. However, there is still a lack of a summary table to which the reader can refer in order to identify the articles that have been reviewed. That is, the references that are central to the article, not those that have allowed the introduction and discussion to be developed. In this table, key and visual details should be summarized: authors, year, type of sample, essential conclusions, etc.
On the other hand, it is convenient to review the bibliography well, many doi are absent.
Author Response
Response to second revision.
Comment:
The authors have made a valuable effort to reorder and make the work more understandable. This is something that is appreciated.
Response: Thank you for your thoughtful review which has led to a general improvement in the readability of the paper.
Comment:
However, there is still a lack of a summary table to which the reader can refer in order to identify the articles that have been reviewed. That is, the references that are central to the article, not those that have allowed the introduction and discussion to be developed. In this table, key and visual details should be summarized: authors, year, type of sample, essential conclusions, etc.
Response: Thank you for your suggestion, the table has been added as Table 1
Comment:
On the other hand, it is convenient to review the bibliography well, many doi are absent.
Response: Thank you for your suggestion, DOIs have been added
